# Sera Protein Signatures of Endometrial Cancer Lymph Node Metastases

**DOI:** 10.3390/ijms23063277

**Published:** 2022-03-18

**Authors:** Doris Mangiaracina Benbrook, James Randolph Sanders Hocker, Katherine Marie Moxley, Jay S. Hanas

**Affiliations:** 1Gynecologic Oncology Section, Department of Obstetrics and Gynecology, Stephenson Cancer Center, University of Oklahoma Health Sciences Center, Oklahoma City, OK 73104, USA; 2Department of Biochemistry and Molecular Biology, University of Oklahoma Health Sciences Center, Oklahoma City, OK 73104, USA; jay-hanas@ouhsc.edu; 3Department of Obstetrics and Gynecology, Rogel Cancer Center, University of Michigan Health System, Ann Arbor, MI 48109, USA; kmoxley@med.umich.edu

**Keywords:** endometrial cancer, lymph node, metastasis, epithelial-to-mesenchymal transition, leave one out cross validation, mass spectrometry, signaling pathways

## Abstract

The presence of lymph node metastases in endometrial cancer patients is a critical factor guiding treatment decisions; however, surgical and imaging methods for their detection are limited by morbidity and inaccuracy. To determine if sera can predict the presence of positive lymph nodes, sera collected from endometrial cancer patients with or without lymph node metastases, and benign gynecology surgical patients (N = 20 per group) were subjected to electron spray ionization mass spectrometry (ES-MS). Peaks that were significantly different among the groups were evaluated by leave one out cross validation (LOOCV) for their ability to differentiation between the groups. Proteins in the peaks were identified by MS/MS of five specimens in each group. Ingenuity Pathway Analysis was used to predict pathways regulated by the protein profiles. LOOCV of sera protein discriminated between each of the group comparisons and predicted positive lymph nodes. Pathways implicated in metastases included loss of PTEN activation and PI3K, AKT and PKA activation, leading to calcium signaling, oxidative phosphorylation and estrogen receptor-induced transcription, leading to platelet activation, epithelial-to-mesenchymal transition and senescence. Upstream activators implicated in these events included neurostimulation and inflammation, activation of G-Protein-Coupled Receptor Gβγ, loss of HER-2 activation and upregulation of the insulin receptor.

## 1. Introduction

Endometrial cancer is one of the ten major cancers in women and its incidence and mortality are increasing worldwide [1,2]. The presence of lymph node metastases in endometrial cancer patients is key to the clinical prediction of disease recurrence and overall survival [3,4,5]. Treatment decisions are based on surgical staging with stages I and II confined to the uterus, while stages III and IV have spread beyond the uterus. Metastasis to the locoregional (pelvic and para-aortic) lymph node basins are classified as stage IIIC1 and IIIC2, respectively. Currently, the method and extent of evaluation for lymph node metastases at the time of surgery is highly controversial [6]. Removal and evaluation of lymph nodes (lymphadenectomy) has the benefit of reducing the potential of hematogenous and lymphatic spread of the cancer to distant sites, but at the cost of increased patient morbidity due to surgical complications such as lymphedema, lymphocyst formation, cellulitis, endothelial and neurovascular injury [6,7]. These issues are especially pertinent to obese and morbidly obese patients in whom surgical resection of the lymph nodes is complicated by poor surgical exposure and increased risk of surgical complications. Additionally, the co-morbidities that result from long-standing obesity often make these women poor surgical candidates [7,8]. Thus, the decision to perform a lymphadenectomy remains a topic of debate among gynecologic oncology surgeons and especially in patients with low grade endometrioid histology and multiple medical co-morbidities who are at low risk of extrauterine disease and high risk of surgical morbidity [9,10].

A less morbid alternative to complete pelvic and aortocaval lymphadenectomy is lymphatic mapping (via imaging following cervical injection with dye) followed by directed sampling of the first lymph node(s) that directly drain the uterus (sentinel lymph nodes); however, the utility of sentinel lymph node mapping especially in high-risk endometrial cancer histologies remains controversial [10,11]. Sentinel lymph node mapping increases the overall detection rate for metastases, but has <5% false negative rate [10] when performed with a complete lymphadenectomy. In clinical trials evaluating the predictive value of sentinel lymph node sampling for endometrial cancer, 67% of gynecologic oncologists performed a back-up lymphadenectomy most notably observed when staging patients at high risk for recurrence [12]. Retrospective, multi-institutional studies found that back-up lymphadenectomy did not improve disease-free or overall survival of high-risk or obese endometrial cancer patients [8,13], but did connote clinical benefit by directing the use of adjuvant therapy in patients with identified positive sentinel lymph nodes. The addition of this therapy, chemotherapy and/or radiation therapy was shown to reduce endometrial cancer recurrence in the pelvic sidewall [14], and expanded the use of surgical staging in high morbidity, obese and medically compromised patients.

Based on the importance of knowing the lymph node status in dictating adjuvant therapy and predicting disease-specific survival from endometrial cancer, the current limitations of radiographic assessments in predicting nodal metastases and the controversies surrounding utilization of sentinel lymph node mapping and biopsy versus complete pelvic and aortocaval lymphadenectomy, additional means to identify and understand the biology of metastatic lymph nodes in endometrial cancer patients are needed. Analysis of peripheral blood represents a novel, low morbidity approach to evaluating tumor characteristics at the time of diagnosis that could provide cancer-specific characteristics that can guide surgical management and adjuvant treatment decisions. The objective of this study was to determine if sera proteomic profiling can prediction the presence of lymph node metastases in endometrial cancer patients and to gain insight into the tumor biology associated with lymphatic dissemination of endometrial cancer. Protein profiles present at significantly different levels in sera collected from endometrial cancer patient with or without lymph node metastases and surgical patients with benign gynecologic conditions were identified and found to predict the presence of metastases.

## 2. Results

Twenty sera samples from each of the following three groups of patients were evaluated by electron-spray ionization MS (ESI-MS), Stages I (N-19) and II (N = 1) endometrial cancer patients (no lymph node metastases), Stage IIIC (N = 19) and IVB (N = 1) endometrial cancer patients and Benign Gynecology surgical patients. All of the endometrial cancer histologies included in the study were endometrioid. There were no significant differences in patient age, body mass indices (BMIs), race, use of tobacco, alcohol, NSAIDs, aspirin, metformin or insulin, or diagnosis of Type 2 diabetes, hypertension, cardiovascular disease or arthritis between the groups (Table 1).

Mass spectra of diluted serum samples were compared between the groups using leave one out cross validation (LOOCV) (Figure 1A). Individual peaks in the electron spray ionization MS (ESI-MS) that exhibited significantly different normalized areas between the different groups were identified (Figure 1B). The area of each peak represents the average amount of observable components detected in the prepared sera at each indicated centroided m/Z for the specific group of patients indicated. The LOOCV analysis of the data were able to discriminate sera of endometrial cancer patients from Benign Gynecology controls (Figure 1C, *p* = 6.6 × 10^−20^). As a validation of the analysis, LOOCV was performed on groups consisting of patients randomly assigned into the groups while matching for age and pathologic staging in each group. If the identification of patients to their randomly assigned group fails (as it did in this situation), this would suggest that the pathology originally defining the patient samples is the major factor producing the group separations. This analysis demonstrated no significant differences in the sera profiles of RND groups (Figure 1D). The overlap of, or lack of differentiation between, the groups when the specimens were randomly assigned to being cancer or benign, provides a negative control for the significant differentiation observed between the groups when they are assigned to their true pathology group of cancer or benign, and suggests that the pathology is the major factor producing the distinction between groups.

LOOCV analysis of the data was also able to differentiate between each combination of two of the three groups (Figure 2A: Stages I and II compared to Benign, Figure 2B: Stages IIC and IV Compared to Benign, Figure 2C: Stages I and II compared to Stages IIIC and IV). Lack of LOOCV analysis differentiation of the groups when specimens were randomized to the different groups instead of being assigned to their true groups documented the lack of arbitrary results (Appendix A. LOOCV analysis of randomized groups). This approach was also able to distinguish sera from ovarian cancer compared to endometrial cancer using ESI-MS data from our previous publication [15] that distinguished sera from ovarian cancer from benign controls, and ovarian cancer without (Stages I and II) and with (Stages III and IV) positive lymph nodes (Figure 2D and Appendix A). Again, there were no differentiation between the groups when LOOCV analysis was performed on samples randomly assigned to groups instead of being assigned to their true groups (Appendix A). LOOCV analysis was also able to distinguish endometrial cancer from ovarian cancer when evaluated by individual stages of I and III (Appendix A). Detailed LOOCV data are provided in Appendix A.

The proteins in these significant peaks were then identified by subjecting five samples from each group (Appendix A) to tandem MS/MS mass peak peptide/protein structure identification. Proteins that are overexpressed in diseased cells can often be detected in blood, considered as biomarkers of the disease conditions, and provide insight into the disease mechanism. Therefore, we evaluated the sera protein profiles of the different groups for their involvement with specific canonical signaling pathways which might be driving the endometrial cancer biology. Ingenuity pathway analysis (IPA) of the differential protein profiles in the endometrial cancer groups predicted multiple canonical pathways to be upregulated or downregulated (Table 2 and Figure 3). Comparison of the Stage I and II and Benign groups predicted repression (negative z-scores) of pathways involved in epithelial-to-mesenchymal transition (EMT), fibrosis, HER-2 signaling, neuroinflammation, GP6 and STAT3; and activation (positive z-scores) of Th2 immune cells, PTEN Signaling and Protein Kinase A (PKA) signaling. Comparison of the Stages IIIC and IV and Benign groups identified activation of multiple pathways involved in neuronal signaling, estrogen receptor (ER) signaling, senescence, oxidative phosphorylation, protein kinase A (PKA), G Coupled Protein Receptor (GCPR) G_βγ_ Signaling and phosphoinositide 3-kinase (PI3K) in B Lymphocytes and Integrin Signaling. No pathways were identified to be significantly repressed in this comparison. The osteoarthritis pathway was the only activated pathway found to be significantly different in the comparison of Stages IIIC and IV and Stages I and II groups.

## 3. Discussion

The results of this study identify proteins that are differentially present in sera from patients undergoing surgery for Stages I and II Endometrial Cancer, Stages IIIC and IV Endometrial Cancer or Benign Gynecologic Conditions. LOOCV analyses of the protein profiles in these groups were able to distinguish the cancer from the controls or ovarian cancer patients, as well as the two different stage groups from each other. These results suggest that sera and novel biomarkers in sera can be used for early diagnosis and management of endometrial cancer. The ability to predict Stages IIIC and IV compared to Stages I and II offers promise for developing a blood-based assay that can be added to the clinical and surgical endpoints currently used to make treatment decisions. These decisions include potentially avoiding lymphadenectomy or sentinel lymph node mapping in patients who are poor surgical candidates. This preliminary finding provides proof-of-principle for the capability of blood-based assays to predict the presence of lymph node metastases in endometrial cancer patients. Further studies with improved mass spectrometry technology and validation of the differential levels of the identified proteins among the three groups of patients are needed to translate these findings toward clinical application.

The results of this study were not likely affected by patient demographics, which were not significantly different between the groups, or by the endometrial histology type, because only the endometrioid type of endometrial cancer histology was included in this study. This is an important consideration as histology type has been classically used to categorize the aggressiveness of endometrial cancer. Before the advent of tumor molecular characterization, endometrial cancer was categorized into a less-aggressive Type I associated with diabetes, obesity, high estrogen exposure and estrogen receptor α expression, and the more-aggressive Type II, which has a higher risk of metastases and occurs more often in post-menopausal non-obese women [16]. Type I endometrial cancers are primarily endometrioid histology, while the more aggressive histologic types of serous, mucinous and clear cell are categorized as Type II endometrial cancers. The Cancer Genome Atlas (TCGA) analysis of endometrial cancer identified four risk categories based on tumor DNA mutation profile subtypes [17]. Although endometrioid histology is not associated with the most aggressive TCGA molecular subtype, it was chosen for this study because it is the most common endometrial cancer histology and allowed availability of a sufficient number of samples within both early- and late-stage patients across a single histologic type. In general, the Stages I and II endometrial cancer group in this study could be considered to be categorized as the less aggressive Type I, while the Stages IIIC and IV could be categorized as the more aggressive Type II endometrial cancer.

The differential proteins identified in this study offer an opportunity to study the similarities and differences of early and late stages of endometrial cancer biological processes. The presence of proteins in blood can reflect altered levels of these proteins within diseased cells. Therefore, the differential proteins in sera of patients with and without positive lymph nodes can provide insight into the biology of the endometrial cancer metastatic process (Figure 4).

In the comparison of the early Stages I and II Endometrial Cancer with the Benign Gynecology Control Group, several processes commonly deregulated in both endometrial cancer and diabetes were identified: EMT, PTEN/PI3K, and STAT3. EMT is a process that drives endometrial cancer metastases conferring on cells the capability to migrate, invade and resist apoptosis, and is considered the first step in endometrial cancer metastases [18]. Animal models have shown that diabetes increases the EMT of cancer cells [19,20]. Consistent with this association, EMT/fibrosis signaling pathways were found in this study to be repressed in Stages I and II, but not in Stages IIIC and IV endometrial cancer. This loss of EMT/fibrosis prevention is concordant with the presence of positive lymph nodes in Stages IIIC and IV, but not in Stages I and II. The pathways that regulate the EMT in cancer cells were also identified to be associated with differential sera proteins identified in this study.

EMT is repressed by PTEN and induced PI3K/Akt pathways [18]. PTEN acts as a tumor suppressor gene and represses activation of the PI3K/Akt pathway [21,22,23]. PI3K and/or PTEN mutations are present in over 90% of endometrioid endometrial cancers [17]. PTEN expression is commonly lost in endometrial pre-cancer and cancer due to genetic and epigenetic causes; however, its association with patient outcome is controversial and its prognostic significance may be modulated by obesity [24]. Upregulation of the PI3K/Akt pathway by hyperinsulinemia has been shown to drive endometrial cancer development [25]. On the other hand, PTEN expression is elevated in diabetes. Elevated serum PTEN levels in women with gestational diabetes has been associated with increased insulin resistance [26]. In this study, activation of PTEN signaling identified in Stages I and II, but not Stages IIIC and IV compared to the Benign groups may be due to the higher association of diabetes with early stage or Type I endometrial cancer compared to late stage or Type II endometrial cancer. Furthermore, the activation of PTEN signaling in the Stages I and II group is only partial, because it is associated with downregulation of three proteins, FGFR1, SOS2 and TGFBR2, which are decreased in PTEN activation, and is counteracted with the upregulation of INSR and MAGI2, which are increased with PTEN signaling.

The PI3K pathway is a major player in diabetes through its induction by insulin and effects on metabolism [27]. PI3K activation and lack of PTEN Signaling activation in the Stages IIIC and IV Group, compared to each of the other groups, is concordant with a loss of PTEN repression of PI3K as endometrial cancer cells undergo EMT and metastasize. The PI3K pathway identified was specific to B lymphocytes and may reflect activation of an immune response. This lack of PTEN activation and presence of PI3K signaling activation in Stages IIIC and IV compared to Benign groups are logical upstream mediators of the activation of oxidative phosphorylation, CA^2+^ signaling pathways and senescence. PTEN inhibits, while PI3K stimulates, oxidative phosphorylation in cancer cells [28]. PI3K activation leads to downstream induction of CA^2+^ signaling through effects on lipids and PKA, which modulate CA^2+^ ion channels [29]. Senescence is induced in cancer cells by several oncogenic events, including loss of PTEN and activation of PI3K signaling, while other alterations mutations compromise this oncogene-induced senescence to allow the cancer to thrive [30].

A pathway common to both Stages I and II and Stages IIIC and IV compared to Benign groups is PKA signaling. The z-score for PKA activation increased from the earlier to the later stages, while only the TTN kinase was present in both the early and late stage PKA pathway proteins. Little is known about the role of PKA signaling in endometrial cancer except for a report that luteinizing hormone increases invasion of endometrial cancer cell lines through activation of PKA [31].

The signaling pathways identified in this study (PTEN, PI3K, PKA, CA^2+^) are known to interact and influence each other’s activities and to mediate the downstream activation of the G_βγ_ GPCR [32]. GPCRs have been shown to be activated by neurotransmitters in ovarian cancer [33]. Neuronal and neuroinflammation pathways were observed to be repressed in Stages I and II and activated in Stages IIIC and IV compared to Benign groups. Proteins involved in neuronal development and excitability have been reported to be overexpressed in endometrial cancer. Axotrophin, which is involved in neuronal development and immune response, was found to be elevated in endometrial cancer and to cause EMT, migration and invasion in endometrial cancer cell lines [34]. HERG K(+)channels, which regulate neuronal and cardiac excitability, were found to be expressed at higher levels in endometrial cancer tissues compared to normal and hyperplastic endometrial epithelium [35]. Neuronal activation pathways may integrate with the other signaling pathways in this study based on their activation of GCPRs in epithelial cancer cells [33].

The HER-2 membrane receptor is another upstream regulator of PTEN/PI3K and PKA signaling identified in this study. Repression of HER-2 signaling was also noted in Stages I and II, but not in Stages IIIC and IV compared to the Benign group. Reports of HER-2 expression association with prognostic significance and clinicopathologic features of endometrial cancer are inconsistent. Positive HER-2 expression has been found to be an independent prognosticator of worse endometrial cancer patient survival in several studies [36,37,38,39,40], while a study of 315 patients with endometrioid histology endometrial cancer found no significant association of HER-2 expression with survival, stages I through IV cancer or other clinicopathologic features [41]. A study of 79 patients with Stages I through IV endometrial cancer with various histologies found that the number of cancers with positive HER-2 staining was greater in tumors with ≤50% compared to ≥50% myometrial invasion and in patients with absence compared to presence of positive lymph nodes [42]. Another study of 68 Stages I and II endometrial cancer also reported the number of cancers with positive HER-2 staining to be greater tumors with ≥50% myometrial invasion compared to ≤50% myometrial invasion [43].

EMT is a downstream consequence of STAT3 activation [44]. In this study, STAT3 signaling was repressed in Stages I and II, but not in Stages IIIC and IV compared to Benign groups. Gain of function STAT3 mutations have been shown to promote diabetes, while inhibition of the JAK kinase upstream of STAT3 has been shown to control lymphoproliferation in a diabetes patient with a STAT3 gain of function mutation [45,46]. STAT3 was reported to be upregulated in early-stage endometrial cancer, and inhibition of its signaling pathway has been shown to repress endometrial cancer stem cells and tumor growth [47,48]. Thus, the STAT3 repression observed in this study is likely counteracted by the significantly elevated levels of the INSR insulin receptor observed in the Stages I and II group compared to benign controls. STAT3 repression in Stages I and II, and the lack of STAT3 repression in Stages IIIC and IV, is concordant with the absence and presence of lymph node metastases, respectively, in these groups.

Platelets are also implicated in the mechanism of metastases in this study based on the repression of GP6 signaling in Stages I and II, and lack of this repression in Stages IIIC and IV. GP6 is a collagen receptor expressed primarily on platelets where it participates platelet activation [49]. Platelets and cancer cells interact in a pathological feed-forward loop in which the cancer cells activate platelets and platelets support cancer cells. Platelet counts are significantly elevated in endometrial cancer patients, and, in combination with other factors, are prognostic for presence of positive lymph nodes and worse prognosis [50,51,52]. Blood-based assays including platelet counts have also been shown to differentiate between endometrial cancer from atypical endometrial hyperplasia [53]. PI3K and CA^2+^ signaling, and oxidative phosphorylation, mediate platelet activation downstream of GP6 [54].

The identification of activation of the osteoarthritis canonical pathway with Stage IIIC and IV compared to Stages I and II endometrial cancers is consistent with some preliminary findings associating arthritis with increased risk of endometrial cancer. There are several case reports of endometrial cancer in patients with arthritis [55,56]. Patients with the arthritis-associated Myhre syndrome were found to have a 9% incidence of endometrial cancer [57]. A retrospective cohort study found a significant elevation of endometrial cancer in patients diagnosed with dermatomyositis, but not with other rheumatoid arthritis conditions (95% CI = 3.7 to 110.3) [58].

## 4. Materials and Methods

### 4.1. Clinical Specimens

The sera samples were obtained from the Stephenson Cancer Center Biospecimen Bank under a University of Oklahoma Health Sciences Center Internal Review Board-approved protocol. The twenty specimens in the Benign Gynecology group were collected from surgical patients with the following pathology diagnoses: ovarian benign neoplasms (N = 8), endometriosis (N = 3), uterine fibroids (N = 2), both ovarian neoplasms and uterine fibroids (N = 3), ovarian and endometrial fibroids and cervical intraepithelial neoplasia (N = 1), atypical endometrial hyperplasia (N = 1), vulvar intraepithelial neoplasia (N = 1) and unknown benign pathology (N = 1). All of the histologies of the 20 endometrial cancers in the Stages I and II groups were endometrioid. All of the histologies of the 20 endometrial cancers in the Stages IIIC and IV groups were endometrioid.

### 4.2. ESI-MS

Serum ESI-MS and mass peak analysis were performed as described [15,59,60,61]. In brief, serum samples were diluted (4 μL sera + 1200 µL of a mixture of 50% methanol, 2% formic acid and 48% water). The ESI-MS was performed on an ADVANTAGE (Thermo Fisher, Waltham, MA, USA) ion-trap mass spectrometer in positive ion mode. The source had been modified with a fused silica tip (Polymicro Technologies: Phoenix, AZ, USA) with a 20 micron inside diameter and a 90 micron external diameter. Voltages were set with 1.75 Kv source voltage and 0.34 µamperes current with a capillary temp of 195 °C, for each injection. The flow rate was 0.23 µL per minute using an Eldex MicroPro HPLC pumping system. Samples were triplicate loop injected for MS spectral data and data were acquired for 15–30 min periods. MS signal data (timed from stable injection peak) were extracted from each file in 1.0 *m*/*z* units.

### 4.3. LOOCV Mass Peak Analysis and Statistics

As previously described [15,59,60,61], post-acquisition MS spectral data processing was performed by locally normalizing each injection data stream set to a total peak intensity value of 100 within a 10 *m*/*z* window along the 350–2000 *m*/*z* observed range. Peaks were identified using valley to valley definition and averaged within closest 1 unit *m*/*z* values. Data acquired in triplicate for each sample were averaged as a representative spectral peak pattern. LOOCV analysis format was used to assess the similarity and significance of peak patterns between the known individual patient groups, using t-tests (significance designated at *p* < 0.05, one-tailed, unequal variance). LOOCV analysis was used to remove the overfitting potential, which is the over-optimistic bias potentially observed when a sample is tested against a set of test variables constructed from a group of samples that also included the sample being tested. Use of the LOOCV procedure reduces this bias by serially removing one sample from the dataset and creating a set of test variables from the remaining samples. The variable sample database was then only valid for use against that particular left-out sample. The sample data were then replaced into its proper group and the next sample is removed to create a new variable database against which it is tested. This sample removal and data base construction continues until all samples have been tested. The peak area at each *m*/*z* was compared to the remaining samples. If the left-out sample peak was above the PCV, then that peak was scored with the pathology of the group with the higher mean at the specific peak; otherwise, the peak was scored with the pathology of the group with the lower mean at that peak. All peaks having been scored, the percentage of peaks scored in each group was plotted on the y axis (see patient scoring distribution figures). Additionally, a database series was created where treatments groups are evenly mixed (referred to as random groups “RND”) to assess the potential for identifying random unrelated data patterns using the same methodology and number of significant peaks. Each sera sample was scored against its respective database by performing the above described LOOCV, analyzing each peak between the 400–2000 *m*/*z* range. Additional detailed information is provided in Appendix A.

### 4.4. Tandem MS/MS and Bioinformatic Analysis

For tandem MS peptide sequence analysis [59], samples from sera were randomly selected (Appendix A) and re-analyzed in the MS ion-trap instrument via selected reaction monitoring MS/MS without chromatographic separation of sera. The mass peaks analyzed were pre-determined in the ESI-MS positive mode and found to be significant from the LOOCV analysis (*p* < 0.05). These peaks were chosen for MS/MS isolation and fragmentation. The significant selected peaks were between a 500–1100 *m*/*z* range and are a master set of the peaks represented in figures showing MS trace peaks. Identification of peak proteins was established using SEQUEST Proteome Discoverer 1.0 (Thermo Fisher) employing the “no cleavage” setting on a Human database created through the Discoverer software from an NCBI non-redundant database downloaded on 6 October 2015. Peptides/proteins MS/MS identification from samples involved a cross-correlation value (Xcorr) of 2.0 or better [62]. For Ingenuity^®^ Pathway Analysis (IPA^®^, QIAGEN, Germantown, MD, USA www.qiagen.com/ingenuity (accessed on 10 January 2022)), associated gene names and the number of identified MS/MS sequences were each imported as base-2 log ratios of untreated tumor sequence “hits” divided by treated tumor “hits”. Detected pathways were manually inspected and verified using Medline/PubMed. Additional detailed information is provided in Appendix A.

### 4.5. Test Metrics

A test/procedure diagnostic value is determined for each group after analysis of each sample through the LOOCV process and is defined by sensitivity, specificity, predictive value and efficiency [63,64]. The sensitivity of the test was determined from TP/(TP + FN) for each comparison, where TP was the number of true positives for known disease presence, and FN was the number of false negatives for known disease presence. Specificity was calculated from TN/(TN + FP), where TN is the number of true negatives and FP is the number of false positives. Each comparison of groups utilized TP, FP, TN and FN values defined using cutoffs #SD (standard deviations) above and below the mean percentage % of classified mass patient serum peaks. The number of SDs from the mean for each group was set equal to each other so there was one cutoff with an equal number of SD on each side from the respective group mean.

## 5. Conclusions

Sera protein profiles have the potential to predict the presence of positive lymph nodes in endometrial cancer patients. The canonical pathways associated with proteins differentially present in sera of Benign Gynecology, Stages I and II and Stages IIIC and IV groups of patients provide insight into the biology of endometrial cancer metastases (Figure 4). Several pathways involved in the Stages I and II versus Benign groups comparison would logically inhibit metastases (activation of PTEN and repression of HER-2, STAT3 and EMT), and concordantly, these pathways were not present in the Stage IIIC and IV versus to Benign groups comparison, suggesting that loss of PTEN activation and HER-2, STAT3 and EMT repression are involved in endometrial cancer metastatic progression. Activation of GPCR, HER-2 and integrin receptors, leading to PI3K and PKA activation, leading to oxidative phosphorylation and CA^2+^ signaling, leading to EMT and senescence, are implicated in endometrial cancer metastasis, along with ER-mediated transcription and loss of PTEN activation. Neurostimulation and inflammation are implicated as upstream mediators of these events.

## Figures and Tables

**Figure 1 ijms-23-03277-f001:**
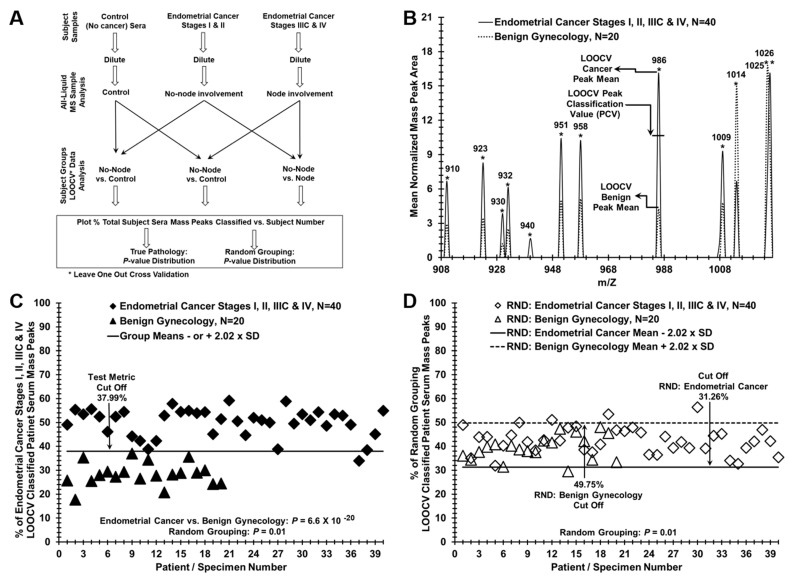
LOOCV of ES-MS data to identify peaks that are significantly different in cancer compared to control subjects. (**A**) Flowchart of approach. (**B**) Plot of peaks (an “*” is above each peak and peak m/z’s are specified) with significantly different areas under the curve/amount of protein detected in cancer and control sera. (**C**) LOOCV of all endometrial cancer stages compared to benign. (**D**) LOOCV of all endometrial cancer stages randomized compare to benign gynecology groups when the specimens are randomly assigned to the groups (RND).

**Figure 2 ijms-23-03277-f002:**
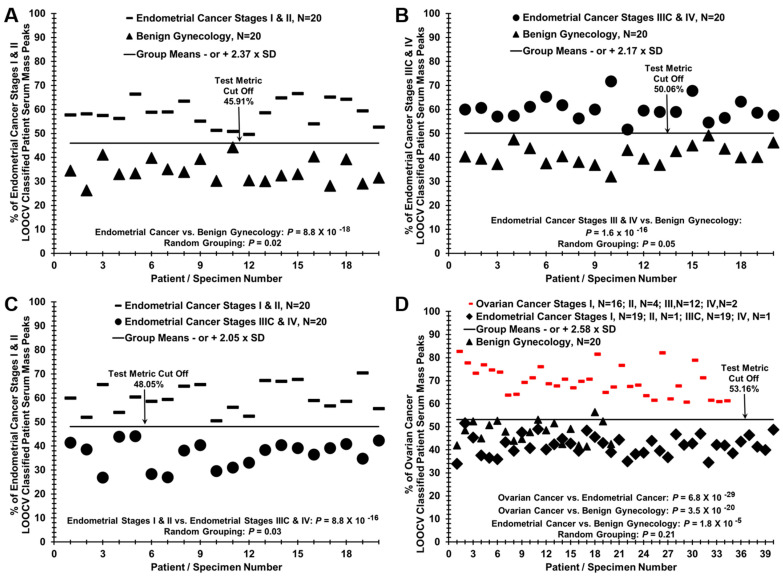
LOOCV analysis comparison of individual groups and ovarian cancer sera. (**A**) Endometrial Cancer Stages I and II compared to Benign; (**B**) Endometrial Cancer Stages IIC and IV compared to Benign; (**C**) Endometrial Cancer Stages I and I compared to Stages IIIC and IV; (**D**) Endometrial Cancer Stages I, II, IIIC and IV endometrial cancer compared to Ovarian Cancer Stages I, II, III and IV.

**Figure 3 ijms-23-03277-f003:**
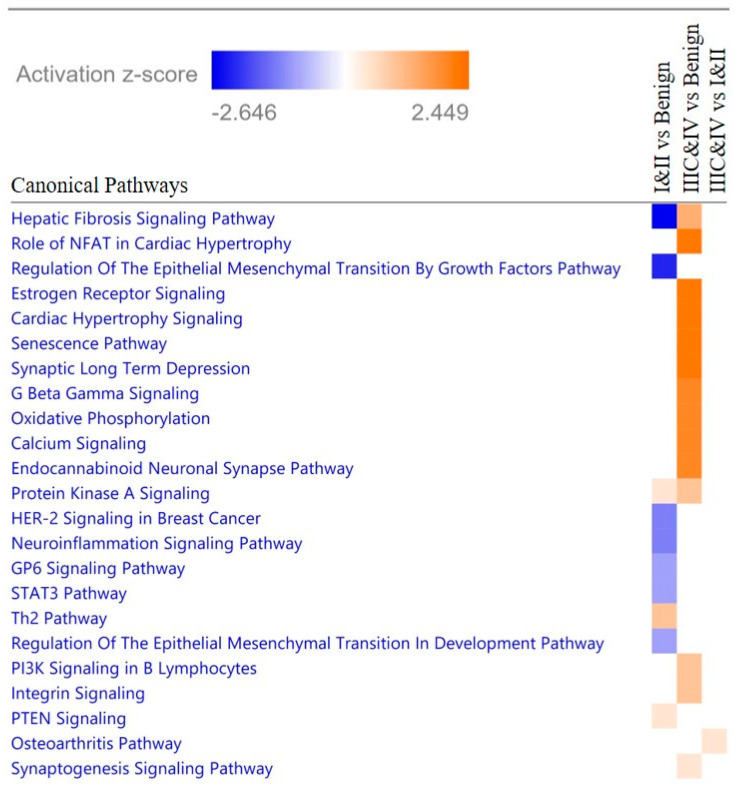
Comparison of z-scores of canonical pathways identified by IPA analysis to be involved in proteins differentially present in sera of groups.

**Figure 4 ijms-23-03277-f004:**
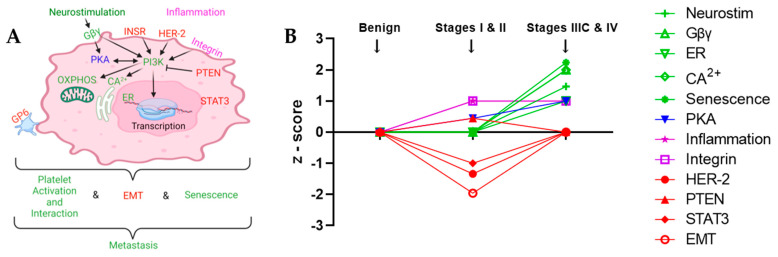
Evolution of canonical pathways in progression from Benign to Early Endometrial Cancer Stages I and II to Metastatic Endometrial Cancer Stages IIIC and IV. (**A**) Illustration of the cellular context of pathway components implicated in endometrial cancer metastases. (**B**) The z-scores for the association of the differential sera proteins in the individual pathways plotted for the different groups. Positive z-scores indicate activation, negative z-scores indicate repression of the pathways. EMT/Fibrosis is the average of hepatic fibrosis signaling, regulation of EMT by growth factors, GP6 signaling and regulation of EMT in development pathways in the Stages I and II group. Neurostimulation (Neurostim) represents the average of cardiac hypertrophy signaling, synaptic long-term depression, role of nfat in cardiac hypertrophy, endocannabinoid neuronal synapse and synaptogenesis signaling in the stages IIIC and IV group. Inflammation refers to Th2 signaling in the stages I and II group and PI3K signaling in B lymphocytes in stages IIIC and IV. (**A**,**B**) In both panels, red indicates loss of pathway regulation is associated with metastatic progression; purple indicates activation of pathway occurs in early endometrial cancer; blue indicates progressively increased activation with metastatic progression; green indicates that pathway regulation is associated with metastatic progression. Subfigure (**A**) created with BioRender.com.

**Table 1 ijms-23-03277-t001:** Comparison of patient characteristics between groups.

Characteristic	Benign	Stage I or II	Stage IIIC or IVB	*p*-Value *
Age, y, median (range)	58 (50–64)	57 (52–63)	62 (32–77)	0.43
BMI, mean (SD)	35.4 (7.9)	35.3 (7.5)	34.7 (6.5)	0.95
Race:				0.56
White	10	17	15	
Black	1	2	4	
American Indian	0	1	0	
Asian	0	0	1	
Unknown/Other	9	0	0	
Endometrioid Histology:				
No	N/A			
Yes	N/A			
Tobacco Use:				
No	15	13	13	>0.99
Yes	1	1	2	
Second-hand	3	6	4	
Unknown	1	0	0	
Alcohol Use:				0.78
No	14	17	16	
Yes	5	3	3	
Unknown	1	0	0	
NSAIDs Use:				>0.99
No	17	14	12	
Yes	3	6	7	
Aspirin Use:				>0.99
No	18	16	16	
Yes	2	4	3	
Metformin Use:				>0.99
No	20	19	18	
Yes	0	1	1	
Insulin Use:				>0.99
No	20	20	18	
Yes	0	0	1	
Type 2 Diabetes:				>0.99
No	19	17	16	
Yes	1	3	3	
Hypertension:				>0.99
No	10	13	10	
Yes	10	7	9	
Cardiovascular Disease:				>0.99
No	19	19	19	
Yes	1	1	0	
Arthritis:				>0.99
No	19	18	14	
Yes	1	2	5	

* Ordinary one-way ANOVA used for age and BMI, Friedman test use for all other characteristics.

**Table 2 ijms-23-03277-t002:** Canonical pathways involved with significantly different proteins between groups.

Group Comparison	Pathway	z-Score	*p*-Value	Molecules *
Stages I and II vs. Benign	Hepatic Fibrosis Signaling	−2.646	0.019	FGFR1, GLI1, ITGB2, NOX1, SOS2, TGFBR2, TTN, WNT6
	Regulation of EMT by Growth Factors	−2.236	0.018	EGF, FGFR1, GSC, SOS2, TGFBR2
	HER−2 Signaling in Breast Cancer	−1.342	0.019	EGF, ITGB2, MT-CO1, NRG1, SOS2
	Neuroinflammation Signaling	−1.342	0.040	CASP8, ELP1, GRIN3B, NOX1, PPP3R1, TGFBR2
	GP6 Signaling	−1	0.018	COL4A3, LAMA2, LAMA5, NOX1
	Regulation of EMT in Development	−1	0.005	GLI1, GSC, JAG2, WNT6
	STAT3	−1	0.022	EGF, FGFR1, INSR, TGFBR2
	Th2	1.0	0.0005	ITGB2, JAG2, NOTCH3, NOTCH4, TGFBR2
	PTEN Signaling	0.447	0.0002	ELP1, FGFR1, INSR, ITGB2, MAGI2, SOS2, TGFBR2
	PKA Signaling	0.447	0.041	AKAP13, PPP3R1, PTPN22, PTPRS, RYR2, TGFBR2, TTN
Stages IIIC and IV vs. Benign	Cardiac Hypertrophy Signaling	2.236	0.01	CACNA1A, CACNA1E, KRAS, NFATC4, PLCG1
	Synaptic Long-term Depression	2.236	0.003	CACNA1A, CACNA1E, GRIA4, KRAS, PLCG1
	Role of NFAT in Cardiac Hypertrophy	2.236	0.006	CACNA1A, CACNA1E, KRAS, NFATC4, PLCG1
	ER Signaling	2.236	0.005	CACNA1A, CACNA1E, KRAS, MT-CYB, MT-ND1, MT-ND5, PLCG1
	Senescence	2.236	0.019	CACNA1A, CACNA1E, EP400, KRAS, NFATC4
	Oxidative Phosphorylation	2	0.003	MT-CO2, MT-CYB, MT-ND1, MT-ND5
	G Beta Gamma Signaling	2	0.005	CACNA1A, CACNA1E, KRAS, PLCG1
	Endocannabinoid Neuronal Synapse	2	0.007	CACNA1A, CACNA1E, GRIA4, PLCG1
	Ca^2+^ Signaling	2	0.026	CACNA1A, CACNA1E, GRIA4, NFATC4
	PKA Signaling	1	0.018	FLNB, NFATC4, PDE3B, PLCG1, PTPN5, TTN
	PI3K Signaling in B Lymphocytes	1	0.007	CBL, KRAS, NFATC4, PLCG1
	Integrin Signaling	1	0.025	KRAS, PLCG1, TLN2, TTN
	Synaptogenesis Signaling	0.447	0.023	CDH23, GRIA4, KRAS, PLCG1, RELN
	Osteoarthritis	0.447	0.0006	FN1, ITGB6, LRP1, NOTCH1, PPARD, TNFRSF1B
Stages IIIC and IV vs. Stages II and III	Osteoarthritis	0.447	0.0006	FN1, ITGB6, LRP1, NOTCH1, PPARD, TNFRSF1B

* Abbreviations defined in Appendix A: Gene IDs and Names of Proteins. Direction of difference between first and second group listed in agreement with pathway activation (green) or repression (red), or no anticipated direction of difference (black).

## Data Availability

https://repository.jpostdb.org/entry/JPST001476 and ftp://ftp.biosciencedbc.jp/archive/jpostrepos/JPST001476 (last accessed 24 February 2022) The following file types have been uploaded: *.raw mass spectrometric (MS and MS/MS) files from ThermoFinnigan/ThermoFisher LCQ-ADVANTAGE, Excel spreadsheets with search results, peak area for analyzed serum, LOOCV results, example LOOCV database file and summarized search tables.

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
