# Peer review of "Sera Protein Signatures of Endometrial Cancer Lymph Node Metastases"

_ijms, 2022, doi:10.3390/ijms23063277_

Round 1

Reviewer 1 Report

I read with great interest the manuscript, which falls within the aim of this Journal. In my honest opinion, the topic is interesting enough to attract the readers’ attention. Nevertheless, authors should clarify some points and improve the discussion, as suggested below.

Authors should consider the following recommendations:

  • Manuscript should be further revised in order to correct some typos and improve style.
  • Accumulating evidence suggests the potential use of novel biomarkers for early diagnosis and management of endometrial cancer. Authors may stress this point, referring to: PMID: 32462419; PMID: 31651558.

Author Response

The manuscript was carefully reviewed and corrected for typos and mistakes.  The following sentence was added to lines 189-190:

These results suggest that sera and novel biomarkers in sera can be used for early diagnosis and management of endometrial cancer.

Reviewer 2 Report

The topic is interesting and the relevance of the article is due to:

  • In endometrial cancer nodal status is important in dictating adjuvant therapies and predicting survival
  • Lymphadenectomy reduces the potential of distant spread, but with an increase of complications, in particular in patients with known multiple medical morbidities
  • The role of lymph node sentinel is still controversial
  • Alternative methods to evaluate the lymph nodes status could be very usefull, in particular this study has the aim to determine if sera proteomic profiling can prediction the presence of lymph node metastases in endometrial cancer patients and to gain insight into the tumor biology associated with lymphatic dissemination of endometrial cancer. It is an interesting proof of principle, that requires further studies.

Below my suggestions:

Manuscript:

  1. The manuscript is well written, however I suggest to better explain figures 1B and 1D, for an easier comprehension also for clinicians not employed in laboratory:
  • for fig 1B what does exactly “different size” mean?
  • For fig 1D why in this case cancer and benign disease are at the same level? Could you please better explain this figure? What does RND mean? (The same for supplemental figures S1 C-D, S2 C-D)
  1. Line 202-203: Please, explain this sentence: why stage I-II and III-IV can be considered type I and II endometrial cancer?
  2. Line 303: please, complete the first sentence.
  3. Line 339: probably it refers to ovarian benign neoplasms
  4. Appendix A, File A4 is not available

Table 1:

  1. Column 2 (Benign): please, verify the number reported for race (the total is not 20)
  2. Column 4 (Stage IIIC or IVB): please verify the numbers, since the total for each characteristic is 19 (not 20)

Author Response

Reviewer 2

The topic is interesting and the relevance of the article is due to:

  • In endometrial cancer nodal status is important in dictating adjuvant therapies and predicting survival
  • Lymphadenectomy reduces the potential of distant spread, but with an increase of complications, in particular in patients with known multiple medical morbidities
  • The role of lymph node sentinel is still controversial
  • Alternative methods to evaluate the lymph nodes status could be very usefull, in particular this study has the aim to determine if sera proteomic profiling can prediction the presence of lymph node metastases in endometrial cancer patients and to gain insight into the tumor biology associated with lymphatic dissemination of endometrial cancer. It is an interesting proof of principle, that requires further studies.

Below my suggestions:

Manuscript:

  1. The manuscript is well written, however I suggest to better explain figures 1B and 1D, for an easier comprehension also for clinicians not employed in laboratory:

Response: The following text was added on lines 102-105 to provide greater clarity:

The area of each peak represents the average amount of observable components detected in the prepared sera at each indicated centroided m/Z for the specific group of patients indicated.

  • for fig 1B what does exactly “different size” mean?

Response: The legend of Figure 1B has been changed by replacing the words “different size” with “different areas under the curve/amount of protein detected”.

  • For fig 1D why in this case cancer and benign disease are at the same level? Could you please better explain this figure?

Response: To further clarify, the following wording has been added to lines 108 – 117:

If the identification of patients to their randomly assigned group fails (as it did in this situation), this would suggest that the pathology originally defining the patient samples is the major factor producing the group separations. This analysis demonstrated no significant differences in the sera profiles of RND groups (Figure 1D). The overlap of, or lack of differentiation between, the groups when the specimens were randomly assigned to being cancer or benign, provides a negative control for the significant differentiation observed between the groups when they are assigned to their true pathology group of cancer or benign, and suggests that the pathology is the major factor producing the distinction between groups. 

  • What does RND mean?

Response: We have defined RND as randomly assigned in Figure Legend 1.

  • (The same for supplemental figures S1 C-D, S2 C-D)

Response: The text on lines 128 to 137 was expanded to provide more clarification and the figure legends for each of these figures have been adjusted to define RND.

  1. Line 202-203: Please, explain this sentence: why stage I-II and III-IV can be considered type I and II endometrial cancer?

Response: This sentence is now on lines 215-218 and has been expanded and reworded to provide clarification as follows:

In general, the Stages I & II and Stages IIIC & IV endometrial cancer group in this study could be considered to be categorized as the less-aggressive Type I, while the Stages IIIC & IV and could be categorized as the more-aggressive Type II endometrial cancer respectively.

  1. Line 303: please, complete the first sentence.

Response: This sentence, now on line 318 has been completed as follows:

EMT is a down-stream consequence of STAT3 activation [44]. In this study,….

  1. Line 339: probably it refers to ovarian benign neoplasms

The term benign ovarian neoplasms now on line 354 has been corrected to ovarian benign neoplasms.

  1. Appendix A, File A4 is not available

Response:  We apologize for the oversight. File A4 has been added to the supplementary Appendix A.

Table 1:

  1. Column 2 (Benign): please, verify the number reported for race (the total is not 20)

Response:  We appreciate the reviewers detailed attention and identification of this error.  The number 19 has been corrected to the accurate number of 9.

  1. Column 4 (Stage IIIC or IVB): please verify the numbers, since the total for each characteristic is 19 (not 20)

Response: This number is truly 19, because two of the specimens were collected from the same patient.
